# Optimization Analysis of Thermodynamic Characteristics of Serrated Plate-Fin Heat Exchanger

**DOI:** 10.3390/s23084158

**Published:** 2023-04-21

**Authors:** Ying Guan, Liquan Wang, Hongjiang Cui

**Affiliations:** School of Locomotive and Rolling Stock Engineering, Dalian Jiaotong University, Dalian 116028, China; guanying2017@djtu.edu.cn (Y.G.); wlq18042781103@163.com (L.W.)

**Keywords:** plate-fin heat exchanger, serrated fin, genetic algorithm, optimization

## Abstract

This study explores the use of Multi-Objective Genetic Algorithm (MOGA) for thermodynamic characteristics of serrated plate-fin heat exchanger (PFHE) under numerical simulation method. Numerical investigations on the important structural parameters of the serrated fin and the *j* factor and the *f* factor of PFHE are conducted, and the experimental correlations about the *j* factor and the *f* factor are determined by comparing the simulation results with the experimental data. Meanwhile, based on the principle of minimum entropy generation, the thermodynamic analysis of the heat exchanger is investigated, and the optimization calculation is carried out by MOGA. The comparison results between optimized structure and original show that the *j* factor increases by 3.7%, the *f* factor decreases by 7.8%, and the entropy generation number decreases by 31%. From the data point of view, the optimized structure has the most obvious effect on the entropy generation number, which shows that the entropy generation number can be more sensitive to the irreversible changes caused by the structural parameters, and at the same time, the *j* factor is appropriately increased.

## 1. Introduction

With the rapid development of science and technology, energy utilization and environmental protection issues have attracted increasing attention, prompting industries such as aerospace, transportation vehicles, shipping, chemical industry and refrigeration to urgently need more efficient, compact and lightweight heat exchange equipment. Plate-fin heat exchanger (PFHE) is the heat exchanger that can meet this requirement. It is also the most widely used type of heat exchanger in the vehicle engineering industry [1,2,3,4,5].

There are many types of PFHE fins such as corrugated fin, louver fin, perforated fin, serrated fin and pin fins depending on the diverse application [6,7,8,9,10,11,12,13]. The serrated fin is a kind of discontinuous fin whose structure is equivalent to that of the flat fin cut into several short segments which are staggered in the vertical direction to form a series of short and staggered fin flow channels. A large number of studies focus on air or other fluids near normal temperature. Many researches have been performed to carry out empirical correlations in serrated fin surface. The correlations of heat transfer data and of friction data for interrupted plane fins staggered in successive rows were developed by Manson [14]. The friction factor correlation for the offset fin matrix was proposed by Webb and Joshi [15]. The general prediction of the thermal hydraulic performance for plate-fin heat exchanger with offset strip fins was provided by Yang and Li [16]. The correlations based on numerical simulation results were proposed by Kim and Lee [17].

Many studies have been developed on the topic of experimental research on heat exchanger [18,19,20,21,22,23,24,25]. The performance parameters of 21 kinds of aviation aluminum serrated plate-fin were provided by Kays and London in wind tunnel experiments [7]. The heat transfer performance of five kinds of aluminum serrated fin were tested by Mochizuki and Yagi, and the performance prediction correlations for serrated fin channels were worked out on this basis [8]. Over the past decades, the design optimization of PFHEs has become an important topic that has attracted lots of interest [26]. According to the second law of thermodynamics, entropy production is caused by irreversible factors of the process. The heat transfer process is a typical irreversible process. The entropy generation minimization method (EGM) is adopted for the analysis of the thermal performance in processes that needs heat transfer [27]. Emerson Hochsteiner et al. presented an optimization of plate-fin heat exchangers (PFHEs) considering as objective function the minimization of the entropy generation units by Adaptive Differential Evolution with Optional External Archive (JADE) and a novel JADE variant, denominated Tsallis JADE (TJADE) [28]. London presented a convenient method to evaluate the loss of thermodynamic irreversibility in the process of equipment [29]. Thus, the minimum entropy generation unit (EGU) includes the minimization of the irreversibility of the system and is designed to increase heat transfer [30].

The design of heat exchanger includes the determination of several geometric parameters under certain constraints [31,32,33]. Its surface geometry is described by the fin length *l*, height *h*, transverse spacing *s*, and thickness *t*. As the secondary heat transfer surface of PFHE, the fins can effectively increase heat transfer area, improve heat transfer efficiency, strengthen compactness, and increase strength and bearing capacity. In addition, the flow and heat transfer characteristics of PFHEs are closely associated with the secondary heat transfer surface. Therefore, optimization of the parameters of fins is very important for energy saving and cost of heat exchanger. In the past few decades, the design optimization of heat exchanger has attracted the attention of many researchers. Wieting [34], Mochizuki et al. [35], Manglik and Bergles [36], and Huizhu Yang et al. [37] performed studies to develop the empirical correlations in serrated fin surface based on experimental data. Yousefi et al. [38] explored the use of a proposed variant of harmony search algorithm for design optimization of plate-fin heat exchangers.

The aim of the present paper is to provide a systematic design methodology that combines 3D CFD simulation calculation analysis and genetic algorithm methods to investigate the characteristics of heat transfer and pressure drop for serrated fins of PFHE. Firstly, by building parametric modeling of serrated fins and a set of numerical simulation, results are obtained. Based on these, the relationships between the important structural parameters of serrated fins and the heat transfer factor and friction factor are established. Compared with the experimental data, the experimental correlations of heat transfer factor and friction factor of this model are determined. Secondly, from thermodynamic point of view, multi-objective optimization calculation is carried out with genetic algorithm.

## 2. Numerical Calculation

### 2.1. Physical Model

Figure 1 shows a three-dimensional calculation model of serrated fin of PFHE. In the figure, *h*, *s*, *t* and *l,* respectively, represent fin height, pitch, fin thickness and serrated tooth length. According to the actual situation, the fin material is aluminum alloy, one side of the fluid is air, and the other side of the fluid is cooling water.

In order to prevent fluid backflow, a transition length is set before and after the flow direction of the three-dimensional fin model so that the simulation calculation is closer to the actual situation. This length is calculated as follows [39]:
(1)$$L_{e}=0.27Re^{0.51}\mathrm{Pr}D_{h},$$
where *Pr* is Prandtl number. The Reynolds number *Re*, and hydraulic diameter *D_h_* are calculated as follows [40]:(2)$$Re=\frac{u_{c}\cdot D_{h}}{\nu },$$
(3)$$D_{h}=\frac{4lA_{c}}{A}=\frac{4l\left(h-t\right)\left(s-t\right)}{2\left[l\left(h-t\right)+\left(s-t\right)+t\left(h-t\right)\right]+t\left(s-2t\right)},$$
(4)$$u_{in}A_{in}=u_{c}A_{c},$$
where $u_{c}$ is flow velocity in fin channel, $u_{in}$ is flow inlet velocity, *ν* is kinetic viscosity, *A_c_* is cross-sectional area of fin channel, *A_in_* is inlet area of extension.

### 2.2. Heat Transfer

The most important performance evaluation index of heat exchanger is the Colburn factor *j*, which is determined by the basic formula of heat transfer factor [41].
(5)j=NuRePr13,
and
(6)$$Pr=\frac{\mu c_{p}}{\lambda },$$
(7)$$Nu=\frac{h_{c}D_{h}}{\lambda_{f}},$$
(8)$$Re=\frac{\rho uD_{h}}{\mu },$$
where *h_c_* is mean heat transfer coefficient of fin channel, *λ* is thermal conductivity, *λ_f_* is thermal conductivity of fluid, *μ* is dynamic viscosity of fluid, *c_p_* is specific heat, and *u* is velocity of flow. *h_c_* is calculated as follows:(9)$$h_{c}=\frac{1}{\eta_{0}}\frac{1}{\frac{1}{K}-\frac{b}{\lambda_{s}}\frac{A}{2A_{W,\mathrm{cp}}}},$$
where *A_w, cp_* is the wall area of the covered plate, *η*_0_ is surface efficiency of fin channel. Heat transmittance coefficient *K* is determined as follows:(10)$$K=\frac{Q}{A\Delta t_{m}},$$
where the heat transfer amount *Q* is calculated by
(11)$$Q=mc_{p}\left(T_{out}-T_{in}\right).$$

The logarithmic mean differential temperature Δ*t**_m_* is calculated by:(12)$$\Delta t_{m}=\frac{T_{out}-T_{in}}{\ln\left(\frac{T_{w}-T_{in}}{T_{w}-T_{out}}\right)},$$
where *T**_in_* is inlet temperature, *T**_out_* is outlet temperature, *T**_w_* is the wall temperature. The surface efficiency of fin channel *η*_0_ is calculated by
(13)$$\eta_{0}=1-\frac{A_{2}}{A}\left(1-\eta_{f,id}\right),$$
where *A* and *A*_2_ represent the total heat transfer area and secondary heat transfer area, which are expressed as follows [42]:(14)A=2[l(h−t)+l(s−t)+t(h−t)]+t(s−2t),
(15)$$A_{2}=2l\left(h-t\right)+2t\left(h-t\right)+t\left(s-2t\right),$$
where *η**_f,id_* is ideal one-dimensional fin efficiency in fin channel, which is calculated as follows:(16)$$\eta_{f,id}=\frac{\mathrm{th}\left(\frac{1}{2}mh\right)}{\frac{1}{2}mh},$$
(17)$$m=\sqrt{\frac{2h_{c}}{\lambda_{s}t}},$$
where *λ**_s_* is the thermal conductivity of solid.

Another important performance index of the heat exchanger is the friction factor *f* that describes the flow resistance characteristics. The formula is simplified as follows:(18)$$f=\frac{\Delta pD_{h}}{2\rho_{f}u_{}^{2}L},$$
where $\rho_{f}$ is density of fluid.

### 2.3. Mathematical Models

LRN *κ-ε* Model is used in this paper to calculate the heat transfer and flow characteristics of the plate-fin heat exchanger with serrated fins. The Abid method is used for simulation calculation [43]. If the influence of buoyancy on heat transfer is not calculated, its control equation and LRN *κ-ε* Model [44] are as follows:

Continuity equation:(19)$$\frac{\partial \left(\rho {\bar{u}}_{i}\right)}{\partial x_{i}}=0.$$

Momentum equation:(20)$$\frac{\partial \left(\rho {\bar{u}}_{i}\right)}{\partial t}+\frac{\partial \left(\rho {\bar{u}}_{i}{\bar{u}}_{j}\right)}{\partial x_{j}}=-\frac{\partial \bar{p}}{\partial x_{i}}+\frac{\partial }{\partial x_{j}}\left(\mu \frac{\partial {\bar{u}}_{i}}{\partial x_{j}}-\rho \bar{{u^{\prime }}_{i}{u^{\prime }}_{j}}\right).$$

Energy equation:(21)$$\frac{\partial }{\partial x_{i}}\left(\rho \bar{T}\right)+\frac{\partial }{\partial x_{i}}\left(\rho {\bar{u}}_{i}\bar{T}\right)=\frac{\partial }{\partial x_{i}}\left(\frac{\lambda }{c_{p}}\frac{\partial \bar{T}}{\partial x_{i}}\right).$$

*k* equation:(22)$$\frac{\partial }{\partial x_{j}}\left(\rho {\bar{u}}_{j}k\right)=\frac{\partial }{\partial x_{j}}\left(\left(\mu +\frac{\mu_{t}}{\sigma_{k}}\right)\frac{\partial k}{\partial x_{j}}\right)+\mu_{t}\frac{\partial {\bar{u}}_{i}}{\partial x_{j}}\left(\frac{\partial {\bar{u}}_{i}}{\partial x_{j}}+\frac{\partial {\bar{u}}_{j}}{\partial x_{i}}\right)-\rho \varepsilon .$$

*ε*equation:(23)$$\frac{\partial }{\partial x_{j}}\left(\rho {\bar{u}}_{j}\varepsilon \right)=\frac{\partial }{\partial x_{j}}\left(\left(\mu +\frac{\mu_{t}}{\sigma_{\varepsilon }}\right)\frac{\partial \varepsilon }{\partial x_{j}}\right)+\frac{\varepsilon }{k}\left(c_{1}\left|g_{1}\right|\mu_{t}\frac{\partial u_{i}}{\partial x_{j}}\left(\frac{\partial u_{i}}{\partial x_{j}}+\frac{\partial u_{j}}{\partial x_{i}}\right)-c_{2}\rho \varepsilon \left|g_{2}\right|\right),$$
where *k* is Turbulent kinetic energy, *ε* is Turbulent Dissipation Rate. They constitute a two-equation *k-ε* Model, which is currently the most widely used turbulence model, while LRN *κ-ε* Model modifies the high *Re* number *κ-ε* Model to automatically adapt to regions with different *Re* numbers. *u* is velocity parallel to the wall, *ρ* is fluid density, $c_{\mu }\;c_{1}\;c_{2}\;\sigma_{k}\;\sigma_{\varepsilon }\;g_{1}$ $g_{\mu },g_{2}$ are the coefficients.

μ is laminar viscosity, and $\mu_{t}$ is turbulent viscosity, which is calculated as
(24)$$\mu_{t}=c_{\mu }\left|g_{\mu }\right|\rho \frac{k^{2}}{\varepsilon }.$$

In the Abid method, the values of $c_{\mu }\;c_{1}\;c_{2}\;\sigma_{k}\;\sigma_{\varepsilon }\;g_{1}$ are 0.09, 1.45, 1.83, 1.0, 1.4, 1.0, respectively, and $g_{\mu },g_{2}$ are determined as
(25)$$g_{\mu }=\tan h\left(0.008Re_{y}\right)\left(1+\frac{4}{Re_{t}^{3/4}}\right).$$
(26)$$g_{2}=1-\frac{2}{9}\exp\left(1-\frac{Re_{t}^{2}}{36}\right)\cdot \left[1-\exp\left(\frac{-Re_{y}}{12}\right)\right],$$
where ${Re}_{t}$ is turbulent Reynolds number, and ${Re}_{y}$ is Reynolds number at *y* from the wall.

### 2.4. Grid Generations and Boundary Condition

The structure of serrated fin is more complex than that of flat fin, but its internal shape changes periodically. Hexahedral structured grid is used for grid division [45,46], and the grid diagram is shown in Figure 2. Through grid independence analysis and comprehensive consideration of calculation time, the final number of model grids is determined to be 3.67 million. The grid independence is verified by the pressure difference between the inlet and outlet. As shown in Figure 3, the grid number does not affect the calculation results after 3 million.

In order to adhere to the actual situation, transition sections are added before and after the model to make the fluid distribution in front of the fin inlet more uniform, so the inlet is set as the velocity boundary condition, and the inlet temperature of cold and hot fluid is provided by the actual working conditions. Pressure outlet is set at the outlet to prevent backflow; because the physical model simplifies the fins, the left and right walls are set as periodic boundaries; the heat transfer surface of fluid and solid is set as the fluid–solid coupling surface, and the upper and lower baffles are set as the heat flow density boundary. The fluid working media used for modeling in this section are air and water, and the material of fins and diaphragms is aluminum alloy.

### 2.5. Entropy Generation Analysis

The heat transfer process in the heat exchanger is a typical irreversible process. According to the second law of thermodynamics, the irreversible degree of heat transfer process can be expressed by entropy generation. The main cause of irreversible loss in heat exchanger is to overcome friction resistance in finite temperature difference heat transfer and fluid flow. The sum of the two is the total irreversible loss of heat exchanger. Following the methodology of Bejan [27,47], the rate of entropy generation can be expressed as
(27)$$\dot{S}=C_{\min}\left[\ln\frac{T_{1,o}}{T_{1,i}}+{\left(R/c_{p}\right)}_{1}\ln\left(\frac{P_{1,i}}{P_{1,o}}\right)\right]+C_{\max}\left[\ln\frac{T_{2,o}}{T_{2,i}}+{\left(R/c_{p}\right)}_{2}\ln\left(\frac{P_{2,i}}{P_{2,o}}\right)\right],$$
where *c_p_* is specific heat, subscript *i* refers to inlet, o refers to outlet; *C*_min_, *C*_max_ are the heat capacity rates of the two fluids. Bejan defined the entropy generation number:(28)$$\begin{array}{ll} N_{s}=\frac{\dot{S}}{\dot{m}c_{p}} & =\frac{C_{\min}}{C_{\max}}\ln\left[1+\varepsilon \left(\frac{T_{2,i}}{T_{1,i}}-1\right)\right]+\ln\left[1-\frac{C_{\min}}{C_{\max}}\varepsilon \left(1-\frac{T_{1,i}}{T_{2,i}}\right)\right]\\ & -\frac{C_{\min}}{C_{\max}}{\left(\frac{R}{c_{p}}\right)}_{1}\ln\left[1-{\left(\frac{\Delta P}{P}\right)}_{1}\right]-{\left(\frac{R}{c_{p}}\right)}_{2}\ln\left[1-{\left(\frac{\Delta P}{P}\right)}_{2}\right] \end{array}.$$
where *ε* is efficiency of the heat exchanger which is provided by
(29)$$\varepsilon =\frac{T_{1,o}-T_{1,i}}{T_{2,i}-T_{1,i}}.$$

### 2.6. Analysis of Simulation Results

#### 2.6.1. Comparative Analysis

The fins of the serrated fin of PFHE with the model of 1/8-15.61 in this subsection are calculated as the original model. The heat transfer mode is cross-flow arrangement, and the fin structure parameters are shown in Figure 1, and the specific structure parameter size is shown in Table 1.

In simulation calculation, the entrance boundary is the velocity boundary, and 14 sets of simulation calculations with *Re* numbers from 350 to 7000 are carried out. The simulation results of the *j* factor and *f* factor on air side are compared with the experimental correlations of Wieiting and Kays [34,40]. This is shown in Figure 4, which demonstrates the comparison of *j* factor and *f* factor on fin air side. It can be seen that the maximum relative error of *j* factor between the simulation results and the correlation formula of the Wieiting experiment is 15.6%, and the minimum is 5.4%. The relative error between the simulation results and the correlation formula of the Kays experiment is smaller, and the fitting degree is higher, indicating that the model can describe and calculate the serrated fin more accurately when air is the working medium.

The simulated calculation value of *f* factor is in the laminar flow region with Re ≤ 1000, which is more consistent with the calculated value of Kays experimental correlation formula. The relative error between the calculated value of the Wieiting experimental correlation and the simulation result becomes smaller after entering the turbulent region, even less than 5%, as shown in Figure 5. Moreover, the maximum error is less than 16%. The experimental correlation of *j* factor and *f* factor of Kays are provided by
(30)$$j=0.665Re_{{}_{l}}^{-0.5},$$
(31)$$f=0.44(t/l)+1.328Re_{l}^{-0.5}.$$

Similarly, the simulation results of the water side can be obtained by performing the simulation calculation on the above fin model. Comparing the simulation results with the experimental correlations of Kim [17], it can be seen, from Figure 6, that the relative error between the simulation result of factor *j* and the experimental correlation is less than 20% when it is in the laminar flow region (*Re* < 2000). When it enters the transition region, the relative error gradually increases, indicating that when this correlation is used for the calculation of such fins on water side, the best application range of Reynolds number is in the laminar flow region. When *Re* < 3000, the contrast error of the *f* factor on water side is less than 20%, that can be seen from Figure 7. Therefore, the experimental correlations of Kim have high reliability in calculating the factor *f* on water side. The experimental correlation formula for the *j* factor and *f* factor on the outlet side are shown in the following Equations (32) and (33). The advantage of this correlation is that the applicable range of Reynolds numbers from low to high (100 ≤ *Re* ≤ 7000) can meet engineering requirements.
(32)j=exp(1.96)(sh)−0.098(tl)0.235(ts)−0.154Re(0.0634lnRe−1.3)(Pr)0.00348,
(33)f=exp(7.91)(sh)−0.159(tl)0.358(ts)−0.033Re(0.126lnRe−2.3).

#### 2.6.2. Nephogram Analysis

As seen in Figure 8 and Figure 9, which show the temperature contour and pressure contour of the fin channel, an obvious temperature boundary layer and pressure gradient can be seen on the surface of each fin and on the front end of the fin, respectively. With the truncation of the fin, the boundary layer shows the periodicity of destruction and re-development on the next fin. In the flow direction, there are very obvious temperature gradients at the front and rear ends of each fin. Seen from the flow direction, serrated fins are like short straight ribs inserted in a straight channel, and these short straight ribs are arranged in a cross periodic manner, which will inevitably break the flow and temperature boundary layer continuously, which is beneficial to heat transfer. Therefore, the geometric size of the fin can significantly change the pressure and velocity distribution in the channel, and the parameters can be optimized through simulation.

## 3. Optimization Method

Using genetic algorithm to solve optimization problems with multiple objectives and constraints is Multi-Objective Genetic Algorithm (MOGA). The height *h*, pitch *l*, spacing *s* and thickness *t* of the serrated fin structure size have great influence on the heat transfer and flow performance of PFHE. Therefore, these four parameters are used as design variables, which is shown by
(34)$$X={\left[h,l,s,t\right]}^{T}.$$

When optimizing the structure of serrated fins, the size range is the constraint condition. Each variable should have a clear upper and lower bound. The specific expression is as follows:(35)$$x_{\min}\leq x\leq x_{\max}.$$

The value range of each variable is as follows:(36)$$\begin{array}{cc} s.t. & \left\{ \begin{array}{l} 3.0\leq h\leq 12\\ 3.0\leq l\leq 9.0\\ 1.0\leq s\leq 5.0\\ 0.1\leq t\leq 0.5\\ T_{2}\leq 56\;{}^{\circ}C \end{array}\right. \end{array}.$$

The serrated fins of plate-fin heat exchanger are optimized from three aspects of heat transfer, resistance and irreversibility. The selection of the objective function is the maximum heat transfer factor *j*, the minimum friction factor *f*, and the minimum entropy generation number *N**_s_*. In addition, the objective function expression is provided by
(37)$$\left\{ \begin{array}{l} F_{1}\left(X\right)=\max j(X)=\max j(h,l,s,t)\\ F_{2}\left(X\right)=\min f(X)=\min f(h,l,s,t)\\ F_{3}\left(X\right)=\min Ns(X)=\max N_{s}(h,l,s,t) \end{array}\right.,$$
and the subprograms for calculating j factor, the *f* factor and *N**_s_* are as follows:

(1) The known parameters of fins, such as inlet temperature, inlet flow and structural parameters are input. (2) Hydraulic diameter is calculated with corresponding fin structure parameters. (3) The heat transfer of the fluid is calculated, then the average temperature is determined on the basis of the outlet temperature known in the test, and the physical property parameters of the fluid are obtained. (4) The fluid flow rate is determined by the optimized Reynolds number, and then the fin width is determined, and the flow area and heat transfer area are obtained. (5) The *j* factor, the *f* factor and *N**_s_* are calculated according to the above formula by using the structural parameters, Reynolds number and physical parameters. (6) The *j* factor, the *f* factor and *N**_s_* are converted into fitness function and the fitness value is calculated. (7) Preferential operation is conducted until the result meets the constraint. (8) Crossover and mutation operations are performed to generate a new population, and the return to step four is realized until the termination condition is met.

## 4. Results and Discussion

### 4.1. The Effect of Fin Configuration Parameters

The structural parameters are changed and the fin model is calculated by CFD method. The simulation results are analyzed as follows.

#### 4.1.1. The Effect of the Fin Height and Fin Spacing

The variation range of fin height *h* is 3 mm, 4.5 mm, 5.5 mm, 7 mm, 9 mm. The variation range of fin spacing *s* is 1.5 mm, 2.62 mm, 3.5 mm, 4.5 mm, 5 mm. The fin tooth length *l* and thickness *t* are maintained at 3.175 mm and 0.102 mm, respectively, and the above dimensions are modeled and calculated, respectively. Based on the water side data, the Reynolds number is 350, and the simulation results are shown in Figure 10 and Figure 11. In these figures, the effect of *h* and *s* on *j* factor and *f* factor can be seen.

It can be seen from Figure 10 and Figure 11 that under the given fin spacing *s*, the *j* factor and *f* factor increase with the increase in fin height. When the fin height is fixed, the *j* factor increases with the increase in fin spacing, while the *f* factor decreases. The increase in *h* can increase the secondary heat transfer area and enhance the heat transfer, while increasing the friction resistance. The increase in the spacing *s* can increase the amount of fluid in the flow space, thus strengthening the heat transfer. At the same time, the increase in fin spacing *s* takes more fluid away from the wall, and the impact of the wall shear stress on the fluid is reduced, leading to the decrease in the flow pressure, and then the *f* factor decreases.

#### 4.1.2. The Effect of the Fin Height and Fin Thickness

The variation range of fin height *h* is the same as above. The variation range of fin thickness *t* is 0.102 mm, 0.2 mm, 0.3 mm, 0.4 mm, 0.5 mm. The fin tooth length *l* and spacing *s* are maintained at 3.175 mm and 2.62 mm, respectively. Other calculation conditions of the model remain unchanged. Figure 12 and Figure 13 show the effect of *h* and *t* on *j* factor and *f* factor, respectively. It can be seen from Figure 12 and Figure 13 that when the fin height is fixed, the *j* factor and the *f* factor increase with the increase in fin thickness, and the increase in fin thickness *t* increases the secondary heat transfer area, thus strengthening the heat transfer. The flow space decreases with increasing fin thickness, thereby increasing the flow resistance and increasing the *f* factor.

### 4.2. Optimization Results and Analysis

The known data of working medium are seen from Table 2. The optimization calculation interface is shown in Figure 14.

The optimization results by MOGA are shown in Table 3. It can be seen that the three objective functions are interrelated. In the process of multi-objective optimization, the change in each structural parameter often causes the objective function to show the opposite change trend. Therefore, multi-objective optimization is actually intended to determine an optimal “compromise point” among these objectives. Determining the optimal solution among many solutions often depends on the mathematical expression of the solution method.

In most cases, there is usually no single optimal solution similar to single-objective optimization in multi-objective optimization, but there is usually a solution set composed of optimal solutions. Table 3 presents some optimal solutions. According to these optimal solutions, relative to the original data, the maximum *j* factor increases by 3.7%, the maximum *f* factor decreases by 7.8%, and the maximum entropy generation number *N**_s_* decreases by 31%. From the data point of view, the optimal structure has the most obvious effect on the entropy generation number *N**_s_*, which shows that the entropy generation number *N**_s_* can be more sensitive to the irreversible changes caused by the structural parameters.

In this paper, firstly, CFD simulation is used to determine the range of structural parameters of PFHE, providing a range of parameter variables for the subsequent MOGA optimization calculation, which makes the optimization calculation more accurate and faster. In order to illustrate the advantages of the calculation results of this method, a comparison is made with the calculation results of the optimization methods in the literature [28], in which methods such as GA (Genetic Algorithm), PSO (Particle Swarm Optimization), BA (Bees Algorithm), JADE (Adaptive Differential Evolution with Optional External Archive), and TJADE (Denominated Tsallis JADE) are used to optimize as objective function the minimization of the entropy generation numbers. The comparison results are listed in Table 4. Reductions for GA, PSO, BA, JADE and TJADE of 69.69%, 42.09%, 41.41%, 28.40% and 25.10% are compared to the Optimization results 3 obtained in Table 3.

Although multi-objective optimization can select the optimal structural parameters that meet our requirements, the effect of changes of each parameter on these three important objective functions needs to be discussed and analyzed separately. Genetic algorithm is used to study the influence of single structural parameters on the target function of serrated fins. The range of each parameter is shown in Table 5.

As shown in Figure 14, the influence of the change in the fin structure size on the *j* factor is presented.

Firstly, among the four structural parameters, only when the fin tooth length *l* increases, the *j* factor decreases. The fin tooth length *l* decreases from 9 mm to 3.175 mm, and the *j* factor increases by 21.7%. It can be seen that the smaller the tooth length *l*, the more beneficial the heat transfer. Due to reducing the fin length *l*, the number of fin dislocations per unit length increases, which correspondingly increases the disturbance to the fluid, thus increasing the *j* factor. Secondly, the fin height *h* increases from 3 mm to 9 mm, and the *j* factor increases by 11.3%. In theory, the increase in the fin height *h* will increase the secondary heat transfer area, making more fluid enter the channel, thus strengthening the convective heat transfer. However, the increasing trend tends to be stable as the value changes, not that the higher the better. Thirdly, the fin spacing *s* broadens from 1.5 mm to 5 mm, and the *j* factor increases by 7%. When the fin height *h* is fixed, the increase in fin spacing *s* increases the flow of fluid in the channel, which can theoretically improve the heat transfer effect, and the number of fluids away from the wall increases, and there are more opportunities to accelerate the subsequent heat transfer. Fourthly, the fin thickness *t* thickens from 0.102 mm to 0.5 mm, the *j* factor increases by 13.74%. From the above simulation nephogram, the serrated fin form is equivalent to adding a series of periodically arranged turbulent straight ribs in the flow channel. When the thickness of the turbulent straight ribs increases, it inevitably strengthens the disturbance to the fluid, enhances the heat transfer effect, and also indirectly increases the secondary surface area of heat transfer, so the *j* factor increases.

As shown in Figure 15, the influence of the change in the fin structure size on the *f* factor is presented. Among the four structural parameters, the increase in fin height *h* and fin thickness *t* enhances the *f* factor, and the increase of fin spacing *s* and tooth length *l* reduces the *f* factor. When the fin height *h* increases from 3 mm to 9 mm, the *f* factor increases by 19.1%. The increase in fin height correspondingly increases the total flow of fluid in the flow channel and increases the equivalent diameter of the flow channel, which enhances the *f* factor. When the fin thickness *t* increases, the *f* factor enhances by 67.7%. The increase in thickness increases the turbulence thickness of the “spoiler straight rib” in the flow direction, making the turbulence more intense and the *f* more obvious. At the same time, the fin thickness increases, and on the whole, it increases the solid blockage rate in the channel, making the fluid flow resistance greater, thus increasing the *f* factor. Moreover, the increase in the fin thickness obviously can improve the *f* factor more than the increase in the fin height. In addition, the *f* factor decreases with the increase in fin spacing *s* and fin tooth length *l.* The friction factor *f* is reduced by 13.23% and 31.13%, respectively, in the range of above size changes. The expansion of fin spacing increases the flow volume in the flow channel, and more fluid is moved far away from the wall, which weakens the influence of the wall shear stress on the fluid, thus reducing the pressure drop and reducing the *f* factor. The increase in the fin tooth length *l* reduces the number of turbulences in the flow process, and the flow boundary layer can be maintained for a longer distance without being broken, so that the pressure difference before and after is reduced, and finally the *f* factor is reduced. It can be seen that the tooth length *l* has a more obvious impact on the *f* factor.

At the same time, it can be seen from Figure 8 and Figure 9 that although the size of the fin thickness *t* is the smallest among the four parameters, it is the only size in the entire flow channel that conflicts with the fluid front, and its small changes can directly affect the temperature and velocity boundary layer. Because the thickness of the boundary layer itself is very small, convective heat transfer is basically completed within the boundary layer. Among the four parameters, only the change in thickness *t* affects the fluid inside the flow passage, while the impact of other parameters occurs around the flow passage and does not directly reach the interior of the fluid. Therefore, the change in thickness *t* can directly affect the heat transfer factor *j*, and similarly, an increase in fin thickness can significantly increase the resistance of fluid flow in the flow passage, thereby increasing the resistance *f* factor.

As shown in Figure 16, with the increase in fin height *h* and fin thickness *t,* the entropy generation number *N**_s_* decreases by 10.4% and 38.5%, respectively. Within the variation range of fin spacing *s*, the entropy generation number increases by 83.4%, and the fin spacing *s* is the most influential parameter among the four structural parameters.

The entropy generation number *N**_s_* also increases significantly with the increase in fin tooth length *l* by about 62.1%. It can be seen from the aforementioned three-dimensional simulation that when the tooth length *l* increases, the *j* factor and *f* factor decrease. Thus, the heat transfer entropy increases and can be deduced from the theoretical formula of entropy generation number. This law is consistent with the theoretical formula analysis. The broadening of fin spacing *s* increases the *j* factor and reduces the *f* factor, but it also increases the total entropy generation number. This shows that in the multi-objective optimization, the calculation of entropy generation considers both the heat transfer entropy generation and the resistance entropy generation. Although we know that the entropy generation caused by the viscous resistance of liquid convection heat transfer process can be almost ignored compared with the entropy generation caused by heat transfer, according to the optimization basis of Bejan’s minimum entropy production rule, the best point between heat transfer and flow resistance can be determined, and the total entropy generation number at this point is the lowest.

## 5. Conclusions

In this paper, the numerical simulation method is used to simulate and verify the serrated plate-fin heat exchanger (PFHE). On the foundation of the minimization of entropy generation numbers, MOGA is run to obtain the optimal structure of serrated fin. The main findings are summarized as follows:(1)In the low Reynolds number region on the air side, the simulation results are more consistent with Kays’s experimental correlation. The experimental correlations of Kim have high reliability in calculating the factor *f* on water side.(2)Through multi-objective genetic algorithm (MOGA), a group of optimal solutions meeting the requirements is obtained, where the maximum *j* factor increases by 3.7%, the maximum *f* factor decreases by 7.8%, and the maximum entropy generation number *N**_s_* decreases by 31%. The parameters of the original data are the structure size with excellent performance after actual test, so the *j* factor and the *f* factor of the optimization results are not significantly exceeded. However, the change in entropy production numbers is very obvious, which shows that it is very effective to analyze the thermal performance of heat exchanger with entropy production numbers as an index to optimize its structural parameters.(3)The influence of four structural parameters on the *j* factor, the *f* factor and the entropy generation number *N**_s_* are investigated based on the single objective genetic algorithm. The results show that the fin length *l* has the greatest influence on the *j* factor, the fin thickness *t* has the greatest influence on the *f* factor, and the fin length *l* has the greatest influence on entropy yield, which are 21.7%, 67.7% and 62.1%. respectively. This shows that the research method of entropy generation minimization combined with CFD simulation and genetic algorithm can effectively optimize the key structural parameters of heat exchanger, could determine an important entry point and provide a basis for the design of heat exchanger.

## Figures and Tables

**Figure 1 sensors-23-04158-f001:**
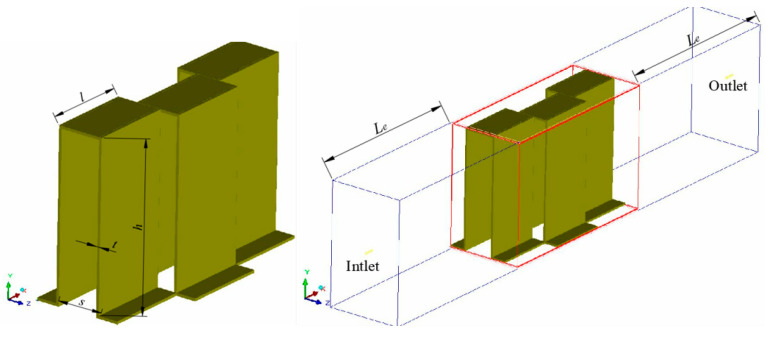
Three-dimensional physical model.

**Figure 2 sensors-23-04158-f002:**
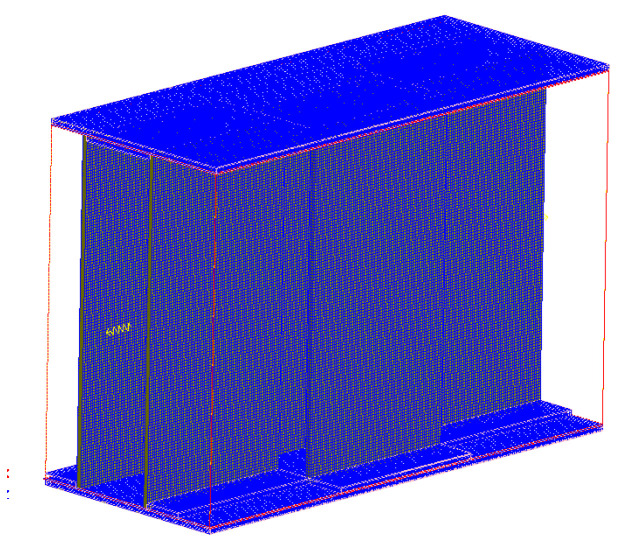
Mesh diagram of model.

**Figure 3 sensors-23-04158-f003:**
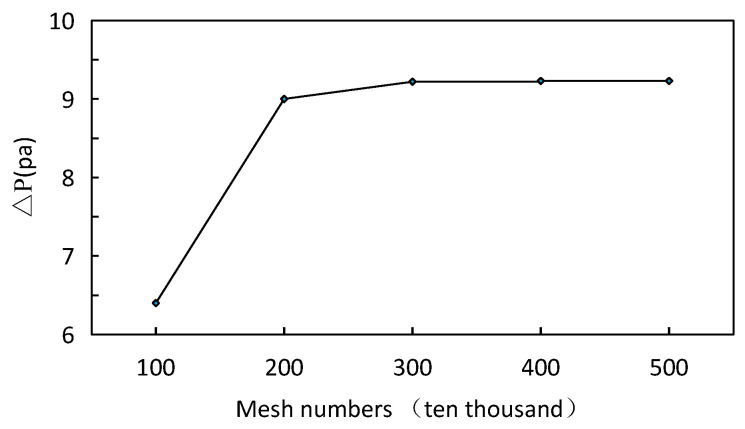
Grid independency check for Scheme.

**Figure 4 sensors-23-04158-f004:**
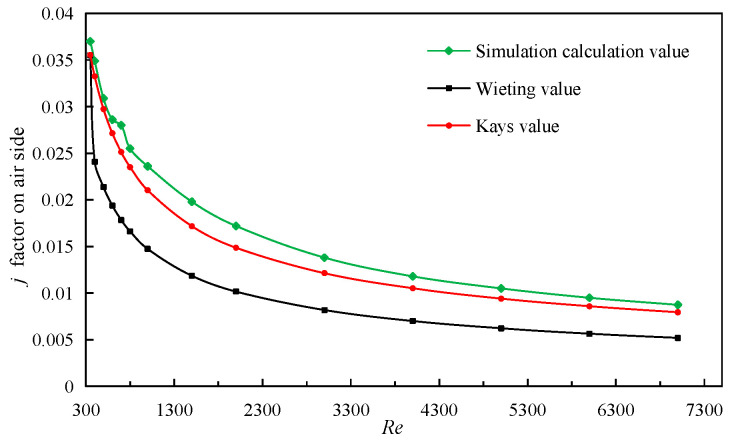
Comparison of *j* factor on air side.

**Figure 5 sensors-23-04158-f005:**
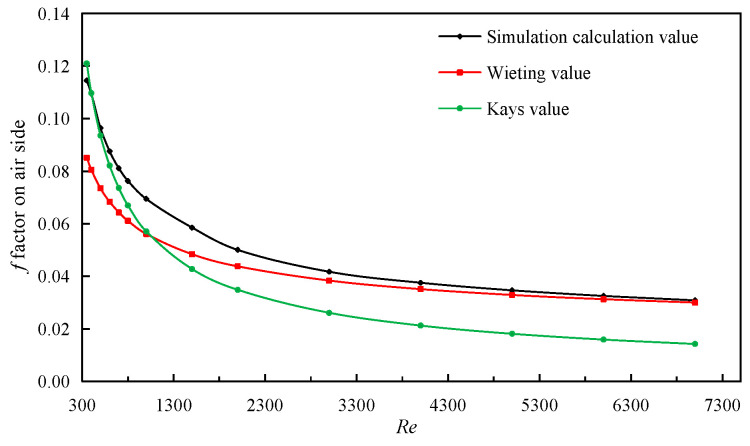
Comparison of *f* factors on air side.

**Figure 6 sensors-23-04158-f006:**
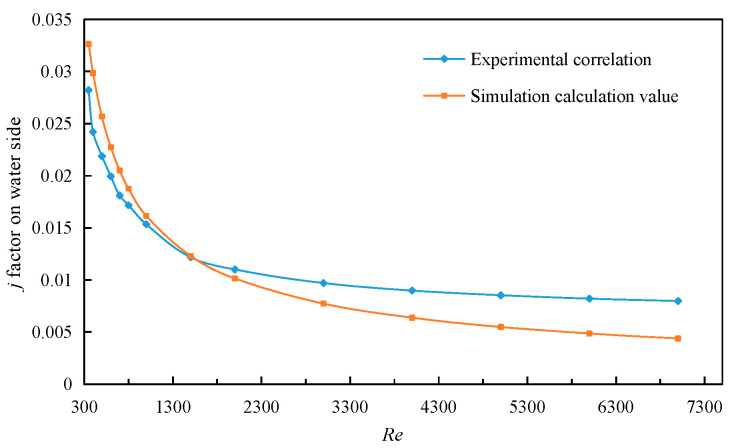
Comparison of *j* factors on water side.

**Figure 7 sensors-23-04158-f007:**
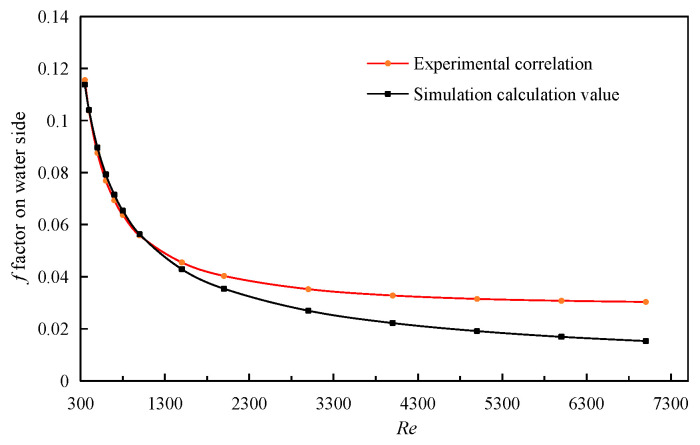
Comparison of *f* factors on water side.

**Figure 8 sensors-23-04158-f008:**
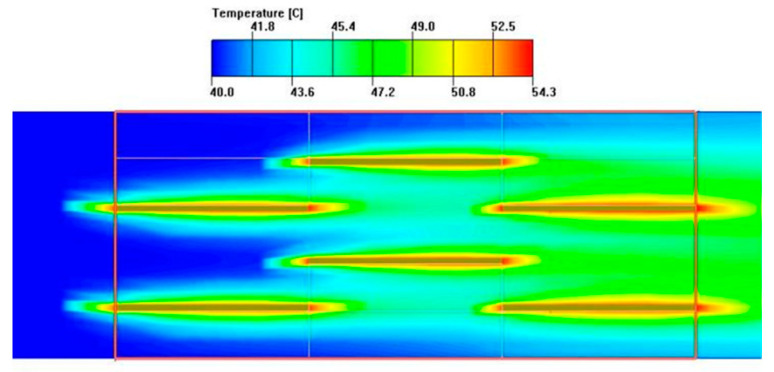
Temperature contour of fin.

**Figure 9 sensors-23-04158-f009:**
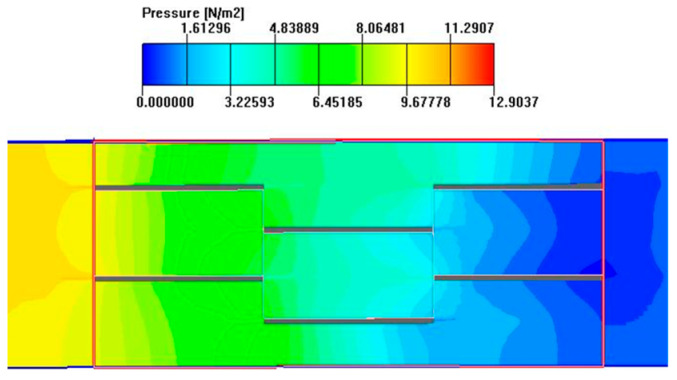
Pressure contour of fin.

**Figure 10 sensors-23-04158-f010:**
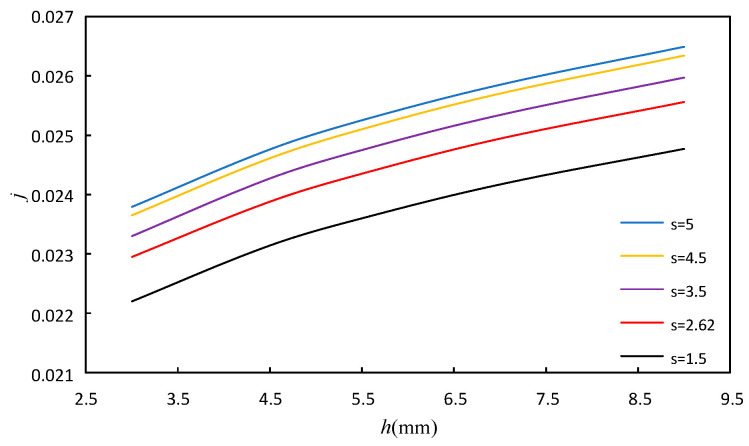
Effect of fin height and fin spacing on *j*.

**Figure 11 sensors-23-04158-f011:**
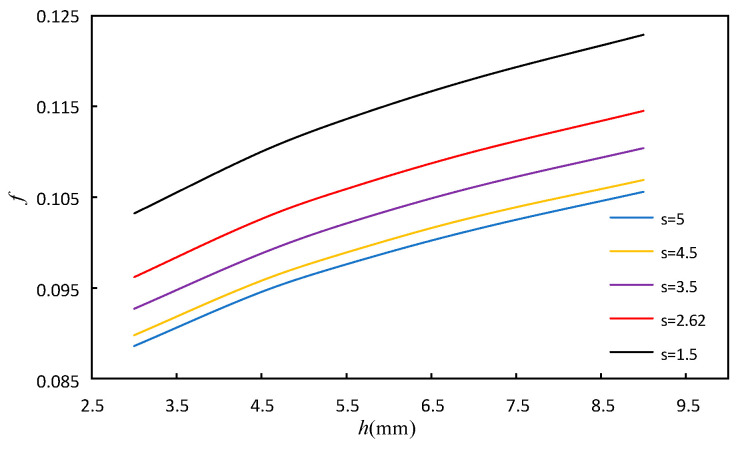
Effect of fin height and fin spacing on *f*.

**Figure 12 sensors-23-04158-f012:**
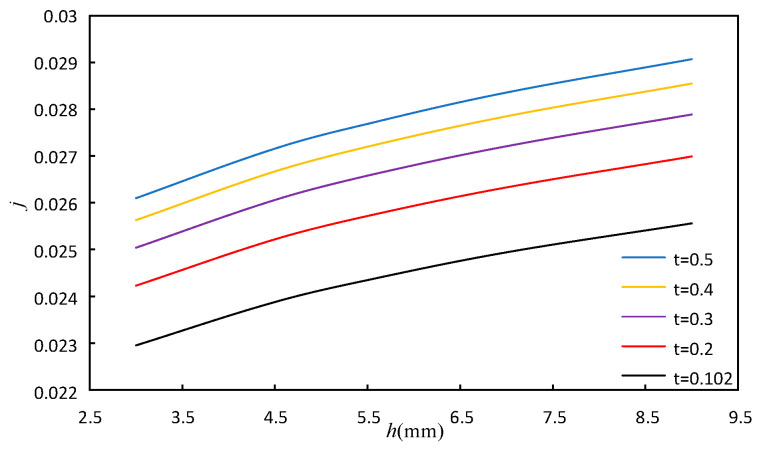
Effect of fin height and fin thickness on *j*.

**Figure 13 sensors-23-04158-f013:**
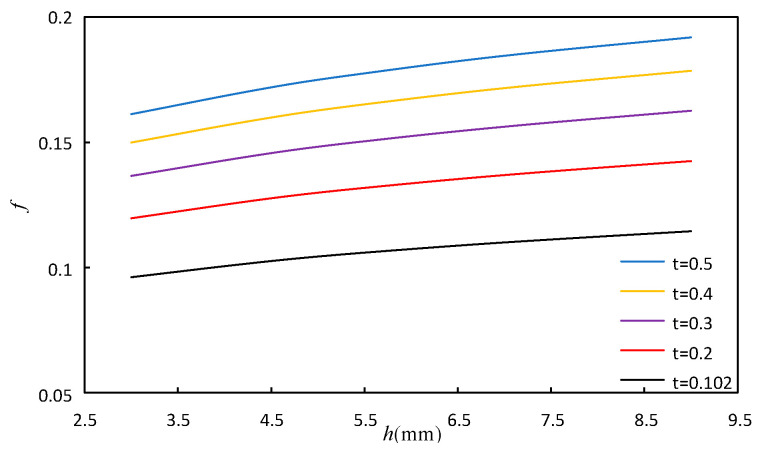
Effect of fin height and fin thickness on *f*.

**Figure 14 sensors-23-04158-f014:**
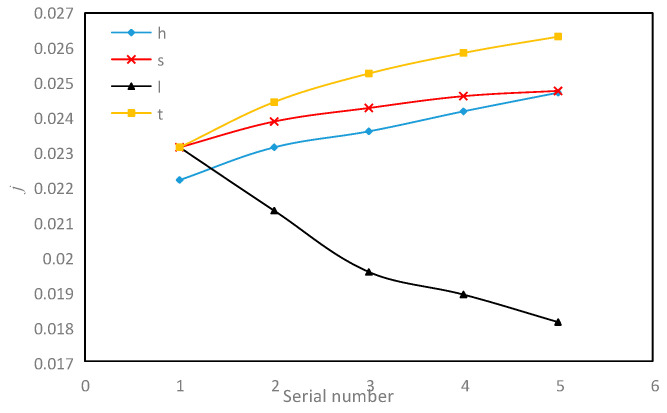
Effect of fin structure parameter on *j* factor.

**Figure 15 sensors-23-04158-f015:**
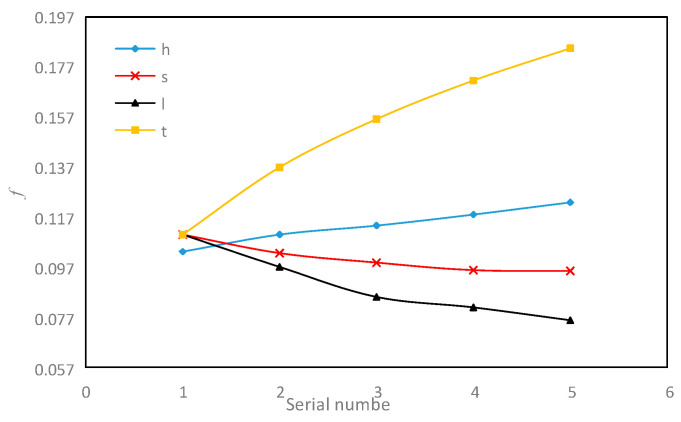
Effect of fin structure parameter on *f* factor.

**Figure 16 sensors-23-04158-f016:**
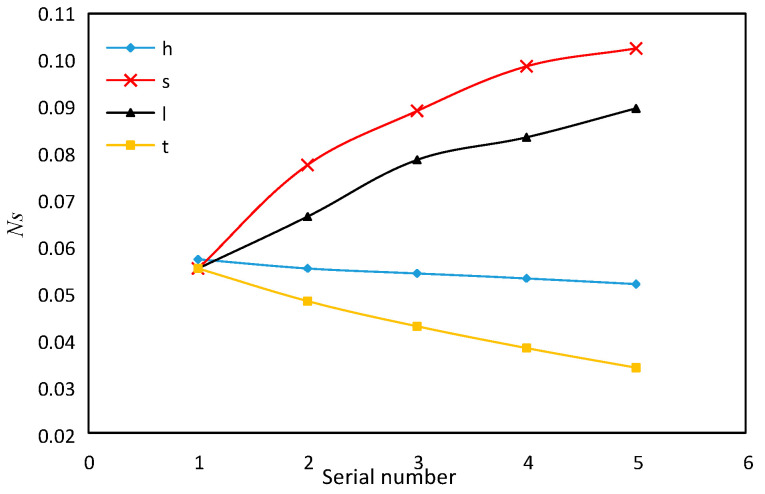
Effect of fin structure parameter on *N**_s_*.

**Table 1 sensors-23-04158-t001:** The original data of the structure size of serrated-fin by simulation.

Fin Structure Parameters	Fin Height *h* (mm)	Fin Spacing *s* (mm)	Fin Tooth Length *l* (mm)	Fin Thickness *t* (mm)
Model size	6.248	1.525	3.175	0.102

**Table 2 sensors-23-04158-t002:** Known data of working medium.

Parameters	Hot Fluid (Water)	Cold Fluid (Air)
Inlet temperature *T*_1_ (°C)	62.6	40
Outlet temperature *T*_2_ (°C)	55.5	55.8
Mass flow *m* (kg/s)	3.28	6.15
Density *ρ* (kg/m^3^)	983.2	1.128
Specific heat *c_p_* (J/kg·K)	4179	1005
Dynamic viscosity *μ* (kg/m·s)	469.9 × 10^−6^	19.1 × 10^−6^
Inlet pressure *P* (MPa)	--	0.11
Thermal conductivity *λ* (W/m·K)	56.94 × 10^−2^	2.496 × 10^−2^

**Table 3 sensors-23-04158-t003:** Comparison of optimization results under objective function.

	Structural Parameters (mm)				Objective Function 1 *	Objective Function 2 *	Objective Function 3 *
Design variable	*h*	*l*	*s*	*t*	*j* _max_	*f* _min_	*N*s_min_
Original data	6.248	3.175	1.525	0.102	0.0239	0.1197	0.05430
Optimization results1	6.808	3.05	1.502	0.102	0.0243	0.1192	0.05213
Optimization results2	2.805	2.18	1.530	0.1	0.0241	0.1157	0.04646
Optimization results3	7.550	2.80	1.02	0.1	0.0245	0.1103	0.03732
Maximum value compared to original data					3.7%	7.8%	31%

* the Objective Function 1, 2, 3 expression are provided by Equation (37).

**Table 4 sensors-23-04158-t004:** Comparison of optimization results with other literature.

	GA	PSO	BA	JADE	TJADE	Results 3 * of This Paper
*h* (mm)	9.53	9.8	9.99	9.99	9.99	7.55
*l* (mm)	6.3	9.8	9.998	8.10	8.82	2.8
*s* (mm)	1.87	2.26	2.46	1.0	1.0	1.02
*t* (mm)	0.146	0.1	0.167	0.101	0.1	0.1
*N_s_*	0.063332	0.053028	0.052886	0.047919	0.046688	0.03732

* Results 3 is shown in Table 3.

**Table 5 sensors-23-04158-t005:** Fin structure parameters of calculation value.

Serial Number	Fin Height *h* (mm)	Fin Spacing *s* (mm)	Fin Tooth Length *l* (mm)	Fin Thickness *t* (mm)
1	3.0	1.5	3.175	0.102
2	4.5	2.62	4.5	0.2
3	5.5	3.5	6.5	0.3
4	7.0	4.5	7.5	0.4
5	9.0	5.0	9.0	0.5

## Data Availability

Not applicable.

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
