# Peer review of "Optimization Analysis of Thermodynamic Characteristics of Serrated Plate-Fin Heat Exchanger"

_sensors, 2023, doi:10.3390/s23084158_

Round 1
Reviewer 1 Report
In this paper, the thermodynamic characteristics of serrated plate-fin heat exchanger are analyzed by using numerical simulation method and Multi-Objective Genetic Algorithm. It has some reference values in engineering. But the following points should be addressed when revising the paper.
(1) The fin thickness t is very small (0.1-0.5mm), but its effect on the factors j and f is so strong as to even larger than that by the fin height h. This is strange and the explanation in the paper seems not so convincible.
(2) To have more values in engineering, it is suggested to give the correlation equations about the factors j and f as a function of structural parameters and flow conditions.
(3) There are some grammar errors, the English writing should be improved.
Author Response
Dear professor,
Please see the attachment.
Yours sincerely,
Regards,
Authors

Reviewer 2 Report
This work investigates the thermodynamic characteristics of serrated plate-fin heat exchangers (PFHE) are analyzed by combining the numerical simulation method and Multi-Objective Genetic Algorithm(MOGA). The work is interesting and is according to the journal's scope. I recommend it for publication after incorporating the following minor corrections.
1. The results obtained are stated in the abstract, however, a comparative study may be provided in this section to show the efficiency of the implemented technique.
2. It will be very interesting if the results are compared with available literature (if present).
3. The references need to be updated from 2019-2023.
4. Some of the equations need to be properly cited for example see., equations 5-8.
5. k equation and epsilon equations may be properly explained, where the parameters present in tensorial as well as in the constant form.
6. Figures may be properly adjusted, for example, see figure 4.
7. The numerical work needs to be improved from the recent work, for example, Accelerating finite element modeling of heat sinks with parallel processing using FEniCSx and Flow of hybrid CNTs past a rotating sphere subjected to thermal radiation and thermophoretic particle deposition.
8. The square cavity is a hot topic and acts as a benchmark problem. To update the results with available work, add, for example, Entropy optimization and thermal behavior of the porous systems considering hybrid and Ferrofluid treatment with an electric field inside a porous cavity considering forced convection.
9. Minor speel check is required for the whole article.
10. The graphs may be explained form its physical point view.
Author Response

(The authors gave the same response as above.)

Round 2
Reviewer 1 Report
All the review comments were addressed. It is suggested to accept the paper in this form.
Author Response
Dear professor,
Thank you very much for the insightful comments. Thank you for your approval of the revised paper. We also checked the English spelling of the paper and some other details in the paper and made further modifications.
Yours sincerely,
Regards,
Authors